# A Regional Approach to Strengthening the Implementation of Sustainable Antimicrobial Stewardship Programs in Five Countries in East, Central, and Southern Africa

**DOI:** 10.3390/antibiotics14030266

**Published:** 2025-03-05

**Authors:** Evelyn Wesangula, Joseph Yamweka Chizimu, Siana Mapunjo, Steward Mudenda, Jeremiah Seni, Collins Mitambo, Kaunda Yamba, Misbah Gashegu, Aquino Nhantumbo, Emiliana Francis, Nyambura Moremi, Henry Athiany, Martin Matu

**Affiliations:** 1East Central and Southern Africa Health Community, Arusha P.O. Box 1009, Tanzania; mmatu@ecsahc.org; 2Zambia National Public Health Institute, Antimicrobial Resistance Coordinating Committee, Lusaka 10101, Zambia; chizimuyjoseph@yahoo.com (J.Y.C.); freshsteward@gmail.com (S.M.); 3Antimicrobial Resistance Coordinating Committee, Ministry of Health, Dodoma P.O. Box 743, Tanzania; smapunjo@yahoo.com (S.M.); emmyfra@yahoo.com (E.F.); 4Department of Microbiology and Immunology, Weill-Bugando School of Medicine, Catholic University of Health and Allied Sciences, Mwanza P.O. Box 1464, Tanzania; senijj80@gmail.com; 5Antimicrobial Resistance Coordinating Committee, Ministry of Health, Lilongwe P.O. Box 30377, Malawi; cmitambo@gmail.com; 6ReAct Africa, Lusaka 10101, Zambia; kaundayamba@gmail.com; 7Rwanda Biomedical Center, Kigali P.O. Box 7162, Rwanda; misbah.gashegu@rbc.gov.rw; 8National Institute of Health, Maputo 1100, Mozambique; aquino.nhantumbo@ins.gov.mz; 9National Public Health Laboratory, Dar es Salaam P.O. Box 9083, Tanzania; nyamburasogone@gmail.com; 10Department of Statistics and Actuarial Sciences, Jomo Kenyatta University of Agriculture and Technology, Nairobi P.O. Box 62000-00200, Kenya; henry.athiany@jkuat.ac.ke

**Keywords:** antimicrobial stewardship, antimicrobial resistance, core elements of AMS, regional approach, Africa

## Abstract

**Background:** Antimicrobial stewardship (AMS) programs optimize the use of antimicrobials and reduce antimicrobial resistance (AMR). This study evaluated the implementation of AMS programs in Africa using a harmonized regional approach. **Methods:** This was an exploratory cross-sectional study across five countries involving 32 hospitals using an adapted Periodic National and Hospitals Assessment Tool from the World Health Organization (WHO) policy guidance on integrated AMS activities in human health. **Results:** This study found baseline scores for AMS core elements ranging from 34% to 79% at the baseline which improved to 58% to 92% at the endline. At baseline, Drugs and Therapeutics Committee (DTC) functionality in updating facility-specific medicines and medical devices ranged from 58% to 100%, and this ranged from 79 to 100% at endline. Classifying antibiotics by WHO AWaRe, classification ranged from 33% to 83% at baseline and 64% to 100% at endline. Leadership commitment scores were 47% at baseline and 66% at endline. Education and training scores were 42% and 63% at baseline and endline, respectively. Reporting and feedback scores were 34% at baseline and 58% at endline. **Conclusions:** Our study showed that understanding context and standardizing regional stewardship approaches enhanced cross-country learning and improved AMS implementation. Although the challenges in Low- and Middle-Income Countries (LMICs) are similar, they vary by country and can be addressed by strengthening AMS regulatory frameworks and surveillance systems.

## 1. Introduction

Antimicrobial resistance (AMR), defined as the ability of organisms to evade the effects of antimicrobial agents, has posed a significant global public health threat to the treatment of common infectious diseases [1]. The net effect of AMR is increased treatment costs through expensive drugs and prolonged hospital stays in addition to rising morbidity and mortality rates [2,3,4,5,6]. Further, the most severe impact of AMR is felt in low- and middle-income countries (LMICs), where weak health systems that are already overstretched are facing a double tragedy as once-treatable conditions are proving difficult to treat [7,8,9]. The resurge against AMR is placed on antimicrobial stewardship (AMS) programs as a strategy where collective efforts are geared toward the appropriate use of antimicrobial agents [10,11,12,13,14,15]. This is achieved through addressing prescription patterns, preventing infections, and fostering multidisciplinary collaborations to optimize patient outcomes and reduce AMR [12,13].

The execution of AMS in LMICs is a hurdle to consider, owing to unique challenges characterized by resource constraints [16,17,18], the sparse and fragmented literature, limited surveillance systems, and human capacity [19,20,21]. Additionally, access to essential medicines is often limited but also compounded by high poverty levels that influence the choice of antimicrobials [22,23]. Furthermore, the disease burden in LMICs is also high, considering the overcrowding, poor sanitation, lack of access to clean water, low literacy levels, limited access to quality healthcare facilities, and inadequate diagnostic capacity, as well as a low health provider-to-population ratio [24,25]. In such settings, the tendency for self-medication, underdosing, and the use of unregulated medicinal and herbal agents are rampant, in addition to a reliance on cultural and superstitious connotations in health [26,27]. Hence, AMS programs in LMICs require an innovative broad-based approach that incorporates appropriate measures to the existing contextual challenges [27,28].

A review of the AMS interventions applied in LMICs demonstrates that multifaceted actions such as education, training, guideline development, multidisciplinary case-based approaches, and policy advocacy, among others, are effective in combating AMR [29,30,31]. The training and education approach aims to improve awareness among the public and healthcare workers on the importance of rational and appropriate antimicrobial use in addition to the simple but impactful actions that can be taken to derail the AMR pandemic [32,33,34,35]. For example, the use of the WHO Access, Watch, and Reserve (AWaRe) classification of antibiotics is targeted at health providers [36]. The AWaRe classification of antibiotics was established by the WHO to optimize antibiotic use and monitor the success of the implementation of AMS programs to address inappropriate prescribing [37,38,39].

The other approach involves the development and dissemination of AMS policy guidelines, where local and external experts combine efforts to provide evidence-based guidelines that suit local contextual needs, the prevailing epidemiology and antimicrobial agents, resources, and the capacity of healthcare systems in LMICs [40]. The guidelines promote a standardized approach to AMS to leverage consensus implementation and monitoring that matches resource constraints [41]. Additionally, another critical strategy to combat AMR is the establishment of multidisciplinary AMS teams at the national, sub-national, and facility levels, comprising infectious disease specialists, microbiologists, nurses, clinical pharmacists, medical officers, healthcare leaders, and other professionals [29,42,43,44]. The teams have diverse roles depending on their establishment level, but those at the facilities are meant to synthesize national guidelines into simple actionable items for their facilities in order to monitor disease and AMR trends. Comprehensive AMS strategies among LMICs are exemplified by South Africa’s AMR national strategic plan [45], and by similar documents from other countries [46,47,48,49,50,51]. 

The impact of AMS interventions in LMICs has been progressive, but the evidence is limited. Process outcomes such as change in knowledge and practices regarding antimicrobial use are notable [34,52,53,54]. Guidelines restricting the prescription or sale of particular antimicrobials have been effective in some countries [55]. Multifaceted interventions have improved prescribing patterns and use of antibiotics [56,57]. In Kenya, the execution of AMS guidelines led to a decreased proportion of patients operated on without appropriate prophylaxis [58], while in South Africa and Ghana, the introduction of clinical pharmacist experts introduced the disciplined use of antibiotics [31,59]. Consequently, it is not easy to establish a reduction in AMR rates due to limited longitudinal studies in LMICs, though some reports have indicated a decrease in Surgical Site Infections (SSIs) [59,60].

In addition, there is a need to improve coordination across facilities, as the fragmentation of health services across tiers of government and service providers is challenging to the implementation of uniform AMS programs. Other variable lessons gained include the findings that a multifaceted approach is critical in LMICs, where interventions are not only tailored to suit the needs of specific countries but also broad enough to meet the requirements within the broader integrated healthcare system [61]. Moreover, interdisciplinary collaborations have fostered shared responsibility, but wider stakeholder engagement is crucial to maintaining the momentum gained so far [62]. Further, multi-country evaluation strategies would be ideal to compare implementation approaches, process, and outcome measures [63].

Therefore, in this study, we evaluated the pre- and post-program implementation of AMS programs across five African countries based on the modified WHO Periodic National and Healthcare Facility Assessment Tool. 

## 2. Results

The countries’ readiness to implement AMS programs was assessed based on the modified WHO AMS Periodic Assessment Tool, which included the following AMS core elements: Drugs and Therapeutics Committees (DTCs), Infection Control Committees (ICCs), or antimicrobial stewardship (AMS) committees; leadership commitment; accountability and responsibility; AMS actions; education and training; monitoring and surveillance; reporting and feedback (Figure 1).

The average scores across the core elements for the assessed hospitals ranged from 34% to 79% and 63% to 92% for the baseline and endline assessments, respectively. The baseline assessment averages showed low performances in leadership and commitment (47%), education and training (42%), and reporting and feedback (34%) for all five countries. On average, improvement was notable in DTC functionality (21%), accountability and responsibility (20%), AMS actions (22%), education and training (26%), and reporting and feedback (24%). None of the assessed countries reported having a fully functional AMS program when assessed across all the core elements (Figure 1).

### 2.1. Functionality of Drugs and Therapeutics Committees in Surveyed Hospitals

All hospitals across the five countries had DTCs, AMS committees, and ICCs in place at the end of the implementation period. However, in assessing the functionality of the DTCs, not all hospitals assessed had updated the medicines and medical devices list or classified their antimicrobials by the WHO AWaRe categorization as shown in Figure 2. All five countries recorded an improvement in DTC functionality from the baseline to endline of the study (Figure 2). 

### 2.2. Leadership Commitment to Supporting AMS Programs in Surveyed Hospitals

AMS was identified as a priority by hospital management/leadership in all countries, with an average score of 47% at baseline and 66% at endline. The average score for AMS activities being included in the facilities’ annual action plan was 49% at baseline and 68% at endline, with scores ranging from 25% to 81% and 55% to 81% at baseline and endline, respectively. This study found that the average scores for the presence of a mechanism to regularly monitor and measure the implementation of AMS activities were 57% at baseline and 69% at the endline. Average scores of 19% and 43% were recorded at the start and end of the implementation period with regard to dedicated financial support for the AMS action plans. Countries’ scores on financing the plans ranged from 8% to 33% at baseline and 13% to 64% at endline.

### 2.3. Accountability and Responsibility in Surveyed Hospitals

At the end of the implementation period, all the five countries reported having multidisciplinary AMS committees with clear terms of reference (TORs) operating at various levels, from partially to fully implemented, in all the assessed hospitals. Four countries reported an improvement in the AMS committees, the only exception being Mozambique (Figure 3). However, not all the AMS committees reported meeting on a regular basis.

### 2.4. AMS Actions in Surveyed Hospitals

Eighty percent of the hospitals assessed reported having STGs during the endline assessment, recording a 14% increase from the baseline. However, there was no improvement reported in the review of guidelines based on evidence. On the pre-authorization of antibiotics based on the AWaRe classification, baseline scores ranged from 25% to 67% while endline scores ranged from 57% to 92% across the countries (Figure 4). There was a decrease in the percentage of facilities reporting having access to laboratory and imaging services to support AMS interventions. Eighty percent of the hospitals reported having IT services, tally cards, or other inventory control tools available that could be used for data gathering to support AMS activities. At the end of the implementation period, no improvement was noticed in hospitals having a written policy/set of guidelines that required prescribers to document the indication and antibiotics prescribed when compared to the baseline (Figure 4).

### 2.5. Education and Training of Healthcare Workers in Surveyed Hospitals

On average, 56% of assessed hospitals had included education programs on topics such as optimizing antibiotic use, prescribing, dispensing, and administering antibiotics in their staff induction training, and 20% reported offering continuous in-service training or continuous professional development on AMS and infection prevention and control (IPC) to hospital staff as at the end of the implementation period (Figure 5).

### 2.6. Monitoring and Surveillance of Antimicrobial Use and AMR in Surveyed Hospitals

The assessment revealed that the hospitals’ capacities for regular prescription audits and point prevalence surveys to assess the appropriateness of antibiotic prescribing were below 75% at the beginning and end of the implementation period. 

At the end of the implementation period, 68% of the health facilities assessed reported regularly monitoring the quantity and types of antibiotics used; 83% regularly monitored shortages/stock-outs of essential antimicrobials; and 57% had a mechanism to report substandard and falsified medicines and diagnostics. With respect to regularly monitoring antibiotic susceptibility and resistance rates, only 63% reported the capacity to perform AST for a range of key indicator bacteria at the endline compared to 52% at the baseline.

### 2.7. Reporting Feedback Within the Healthcare Facilities

Concerning AMS committees analyzing and reporting on the quantities of antibiotics purchased or prescribed, the scores were 17% to 88% at baseline, while the endline scores ranged from 42% to 88% (Figure 6). At the end of the implementation period, 62% of the assessed hospitals had the capacity to develop, aggregate, and regularly update their antibiograms, compared to 48% at the baseline. Low scores were recorded with respect to linking the reported healthcare-associated infections (HAIs), antimicrobial use, AMR, patient outcomes, and quality of care (baseline scores of 8% to 58% and endline scores of 30% to 65%).

## 3. Discussion

To the best of our knowledge, this is the first study that has utilized a regional approach to strengthening the implementation of sustainable AMS programs in five countries in East, Central, and Southern Africa. 

In this study, the core elements of AMS had suboptimal scores at baseline but scored high at endline, and this was contrary to suboptimal performance across all the core elements of AMS in hospitals in a study performed in Ghana [64]. Our study further found that all the hospitals had DTCs, but their functionality scores were low at baseline and improved at the endline stage after implementation of interventions. The presence of DTCs in studies conducted elsewhere showed remarkable differences across countries, for example, a low functionality of DTCs was shown in Zambia, whereas a lack of AMS programs was reported in some selected facilities in Nigeria [14,65]. A study in Nigeria found that hospitals had DTCs but lacked AMS and IPC committees [65], and it is well known that facilities that lack AMS committees tend to fail to implement AMS programs [43]. IPC committees are important to establish and implement IPC core elements and measures in hospitals [66,67,68,69].

Our study found low leadership commitment, accountability, and responsibility at the baseline stage, which also improved at the endline stage after interventions. These core elements foster a sense of responsibility among healthcare professionals for ensuring adherence to antimicrobial prescribing guidelines and minimizing the development of AMR. Evidence has shown that leadership is responsible for operating AMS programs by providing and prioritizing resources needed for AMS activities [70]. Strong leadership engagement is critical to the effective implementation of AMS programs in hospitals [71].

The present study revealed that AMS actions in the surveyed hospitals improved after the interventions, with hospitals reporting a 14% increase in having STGs, an improvement in implementing the WHO AWaRe classification of antibiotics, and having guidelines that required prescribers to document the indication and antibiotics prescribed in prescriptions at health facilities. These reported improvements reveal the significance of implementing AMS programs in hospitals to promote the rational use of antimicrobials, including fostering conformity to the WHO AWaRe classification of antibiotics [72,73]. Despite variations in the scores across countries, our findings indicate an improvement in AMS actions from the baseline to endline of the survey. This reiterates the need for the continuous strengthening of AMS teams beyond their establishment.

In this study, the education and training of healthcare workers regarding AMS was low during the baseline survey and improved at endline. Studies have demonstrated that education is an important component of AMS interventions and promotes the rational prescribing and use of antimicrobials [32,33,70,74,75]. Education and training using the WHO health workers’ training curriculum and framework is critical in improving awareness and knowledge of AMR, leading to the optimization of antimicrobial use [30]. Further, instigation of training in healthcare facilities is essential to improve the awareness and understanding of AMR among healthcare workers [76,77]. Educational programs for new healthcare workers should include the optimization of antimicrobial use, prescription, dispensing, and administration [30]. Furthermore, hospital staff must enroll in continuing professional development education concerning AMR, AMS, and IPC [30].

The performances in the two core elements of monitoring and surveillance and accountability and responsibilities recorded improvements across all five countries at the end of the implementation period. One country reported full implementation for both core elements, while the other four countries demonstrated that both core AMS interventions were partially implemented and needed attention for strengthening. The performance in accountability and responsibility could possibly be attributed to the inclusion of AMS activities in hospital plans, the establishment of AMS committees with terms of reference (ToRs), and the existence of dedicated champions for AMS activities in most of the assessed hospitals. Although most of the assessed hospitals in these countries had costed action plans, there was no dedicated financial support for AMS interventions. For the sustainable implementation of AMS programs, there is a need to provide financial support for AMS activities in hospitals [14,36,64,78]. This is in line with other reports, which indicated that despite many countries having the NAPs in place, very few succeeded in the implementation stage due to inadequate funding for the activities [79]. Furthermore, variations can also be associated with the enrolment of hospitals in some countries that had no pre-existing structured AMS programs, compared to others which already had pre-existing AMS programs. Lastly, the level of hospitals (secondary versus tertiary) can also account for the variations in the AMS performance obtained across the countries.

The implementation status of AMS programs in the five countries varied. While the countries have made progress in establishing and implementing AMS programs, there are significant challenges in fully operationalizing the programs. The lack of human and financial resources, AMS education and training, antibiograms, and the inadequate enforcement of regulations to promote rational antibiotic prescriptions remain huge barriers to implementing and achieving AMS activities and goals [71,80]. It should be reiterated that antibiograms are useful in fostering appropriate empiric treatment of infections in hospitals [81]. However, unreliable or a lack of antibiograms is a hindrance to the effective implementation of AMS [82]. The DTCs provide an opportunity for a strong and sustainable foundation for the implementation of AMS programs within and across countries by fostering (among other AMS interventions) the generation and utilization of antibiograms to guide rational antimicrobial prescriptions.

### Study Limitations

Since the study was conducted across five countries, the practices and resources in these hospitals may differ and could affect the findings. Therefore, generalization to all African countries should be made cautiously. We are also aware that cross-sectional studies may have limitations due to a lack of generalization, information bias, and difficulty in establishing causality. Due to funding and time limitations, the potential impact of using this model on patient outcomes, behavior change, and the rational use of antimicrobials could not be determined. Additionally, staff attrition in hospitals during the implementation period may have had an impact on feedback during the endline survey. Furthermore, multicountry evaluation strategies and studies would be ideal to compare implementation approaches, processes, and outcome measures. However, this study highlights the importance of instigating AMS programs to combat AMR, as evidenced by the improved scores after the AMS interventions.

## 4. Materials and Methods

### 4.1. Study Design and Site Selection

Exploratory cross-sectional studies were performed at the beginning and the end of the project in five countries across 32 hospitals. The countries included Malawi, Mozambique, Rwanda, Tanzania, and Zambia. The included hospitals were five from Malawi, four from Mozambique, eleven from Rwanda, and six each from Tanzania and Zambia (Table 1 and Figure 7). The aim of the project was to strengthen AMS programs in the five countries under the World Bank-funded Strengthening Pandemic Preparedness Project. The project ran from February 2022 to December 2023. The WHO, US Centers for Disease Control and Prevention (CDC), and ECSA-HC have recommended the Core Elements for AMS to support the establishment and implementation of effective AMS programs [83,84]. The guidance documents further outline the structural and procedural components, competencies, and resources that are associated with successful AMS programs.

The targeted countries were part of the Strengthening Pandemic Preparedness Project. The individual countries, through their Antimicrobial Resistance Coordinating Committees (AMRCCs), were engaged to select the hospitals that were involved in the project. All the countries purposively considered hospitals that had basic microbiological capacities to support AMR surveillance, which included secondary- and tertiary-level hospitals. 

### 4.2. The Approach

Countries developed national guidelines, adapting from global and regional guidance as well as accompanying training packages on establishing AMS programs in healthcare settings, with the subsequent identification of participating hospitals. The selected hospitals were requested to establish AMS committees with clear terms of reference and invited to develop action plans on how they intended to implement AMS activities. This was set up to tailor interventions based on country and healthcare facility contexts. Hence, some of the activities involved the conducting of the baseline and endline surveys. The baseline survey was used to assess initial capacity to better understand the needs and priorities of AMS and guide the development of action plans, while the endline survey was used to review the impact of the activities that were implemented on AMS.

### 4.3. Data Collection Tool

Though there are several tools to assess AMS activities in countries, the countries agreed upon and selected the validated adapted Periodic National and Healthcare Facility Assessment Tool from the WHO policy guidance on integrated AMS activities in human health [14,85]. The tool was modified to capture the necessary components for AMS implementation. 

Using the modified validated Periodic National and Healthcare Facility Assessment Tool from the WHO policy guidance on integrated AMS activities in human health [85], the countries conducted baseline assessments and developed implementation plans based on their findings. They then proceeded with the implementation of the activities and monitored their progress throughout and at project completion. Six WHO AMS core elements and an additional core element on drugs and therapeutic committees were assessed. Each core element had several components of assessment or indicators. These included general (presence of DTCs and IPC and AMS teams), DTC functionality, leadership commitment, accountability and responsibility, AMS actions, education and training, monitoring and surveillance, and reporting and feedback within the hospitals. However, under each core element, the countries agreed on selected indicators that they considered to be key for monitoring 

### 4.4. Baseline Assessment

Face-to-face interviews were conducted by four data collectors per hospital, and the information was verified from existing sources and records in the hospitals. The data collectors were trained in how to collect data and enter it into the assessment tool. The data collectors or assessors were members of the AMRCC in their respective countries. During this process, the ECSA-HC and other experts from the selected countries provided technical assistance and guidance to the AMRCCs. The data collected were entered into tablets and computers, with access strictly limited to authorized personnel. The data collected during this phase were used as reference points for comparison with post-intervention data (Figure 2).

### 4.5. Implementation of the Project

After the gap analysis was conducted at the hospital level in the project countries, context-specific and tailored action plans were developed to address the gaps. For sustainable capacity building, a regional peer-to-peer model was embedded in the capacity building process to facilitate benchmarking and peer-to-peer (country-to-country/site-to-site) learning based on the level of the existing capacity or capability for the different core elements in AMS. The implementation process adopted a stepwise approach (Figure 8) to support the five project countries in building their capacities for AMS programs in the selected facilities. The activities involved document development such as AMS guidelines and training manuals, treatment guidelines, local antibiograms, and the classification of antibiotics according to WHO AWaRe. Other activities included the orientation of hospital leadership on AMS, training of health workers in AMS/AMR, conducting point prevalence surveys, antibiotic prescription audits, analysis, and the use of locally generated data for the review of action plans and decision making. Throughout the implementation phase, a regional approach was followed by the implementing team, led by AMRCC members in each country. The teams monitored the process to ensure interventions were implemented as planned. Peer-to-peer cross-country learning and experience sharing by regional experts for the participating countries was adopted for lesson learning throughout the implementation period. The stepwise process adopted is summarized in Figure 8 and Figure 9.

### 4.6. Post-Intervention Data Collection (Endline Assessment)

Following the implementation of the interventions identified, post-intervention data were collected using similar methods and tools employed during the baseline data collection phase. The data captured the changes resulting from the interventions and provided a basis for evaluating the effectiveness of the regional approaches employed. The findings in the post-intervention phase were used to develop sustainable AMS programs (Figure 9).

### 4.7. Data Analysis

The information collected was entered into the Microsoft Excel WHO self-scoring assessment tool. The tool then generated a summarized score based on the eight core elements of the survey. Subsequently, all the responses were grouped into eight themes under each core element. Following this, thematic analysis was conducted to analyze all the responses thoroughly.

The answers were ranked from 0 to 4, where 0 meant “No”, 1 meant “No, but a priority”, 2 meant “Planned but not started”, 3 meant “Partially implemented”, and 4 meant “Yes (Fully implemented)”. When the total AMS score fell between 0.0 and 49.9%, it meant that the AMS program was either non-functional or operating poorly and that it needed to be given priority care. A score of 50–79.9% meant that the AMS program was only partially operational and required support to be strengthened, whereas a score between 80 and 100% meant that the AMS program was fully established and operating as intended, but it still needed ongoing assistance to be sustainable [14,85].

## 5. Conclusions

This study found suboptimal performance of all hospitals in the surveyed countries regarding the core elements of AMS during the baseline phase. However, there was an improvement in the performance of hospitals across all core elements of AMS after interventions were instigated. The selected countries made efforts to establish AMS programs by implementing national AMS programs that included the development of national guidelines, education, and training approaches for healthcare workers through a stepwise approach to the implementation of AMS programs. These programs required a stepwise approach and flexibility due to the complexity of medical decision making surrounding antibiotic use, the diversity of healthcare system capacities, and the variability in the size and types of care. The overall strengthening of health systems is required for the successful implementation of the AMS programs. Collaboration between countries and international organizations can facilitate the development and implementation of effective AMS programs in sub-Saharan Africa. This is through providing sustainable regional networks, aiding in the development of context-specific regional and national guidance documents, the utilization of regional resources and partnerships to enhance north-to-north lesson learning, and sharing platforms for better health outcomes for patients. 

## Figures and Tables

**Figure 1 antibiotics-14-00266-f001:**
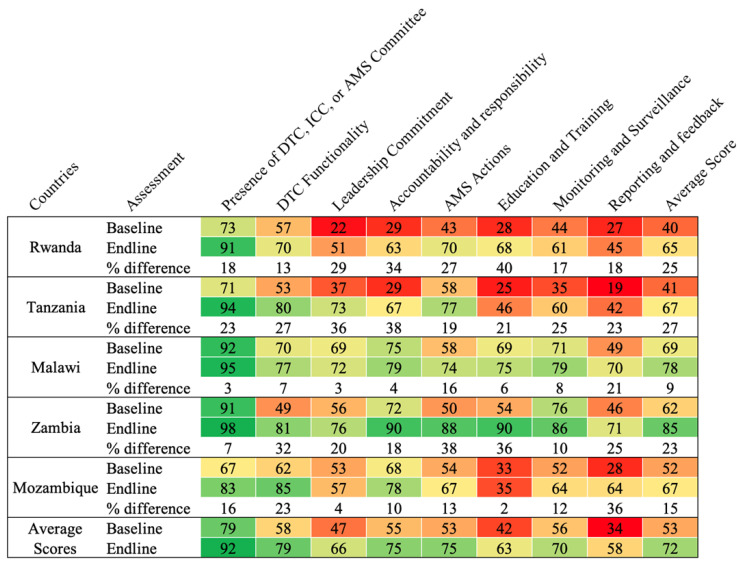
Average performance by country regarding WHO AMS core element indicators. DTC = Drug and Therapeutics Committee; ICC = Infection Control Committee; AMS = antimicrobial stewardship. Colors code: Green = high, Yellow = moderate, and Red = Poor scores.

**Figure 2 antibiotics-14-00266-f002:**
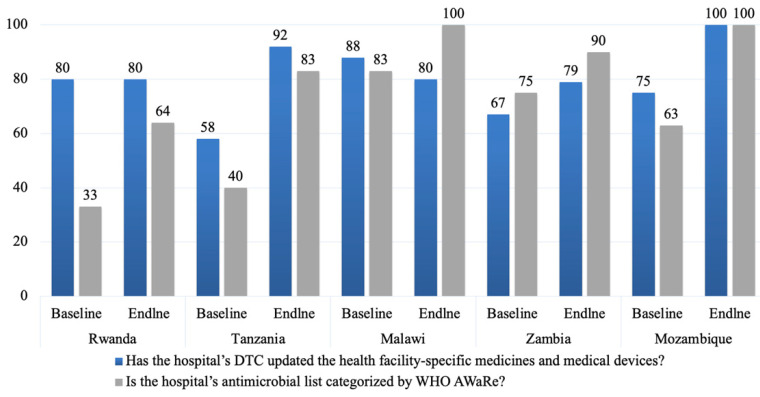
Availability of updated health facility-specific medicines and medical devices and classification of antibiotics by WHO AWaRe categories.

**Figure 3 antibiotics-14-00266-f003:**
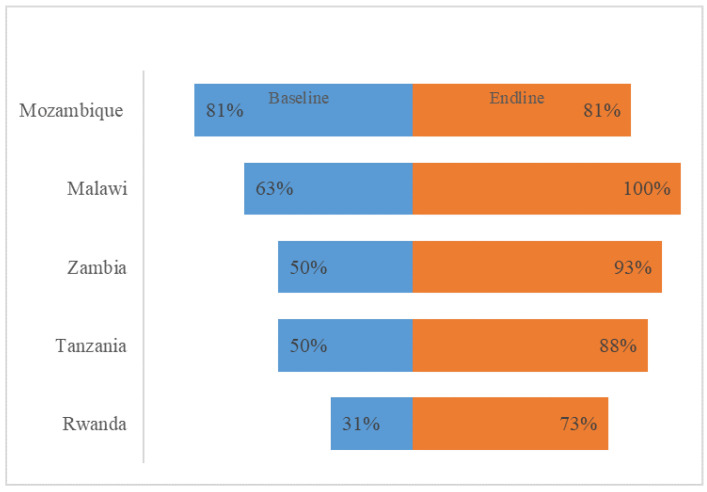
The presence of a multidisciplinary AMS committee in the healthcare facilities with clear terms of reference.

**Figure 4 antibiotics-14-00266-f004:**
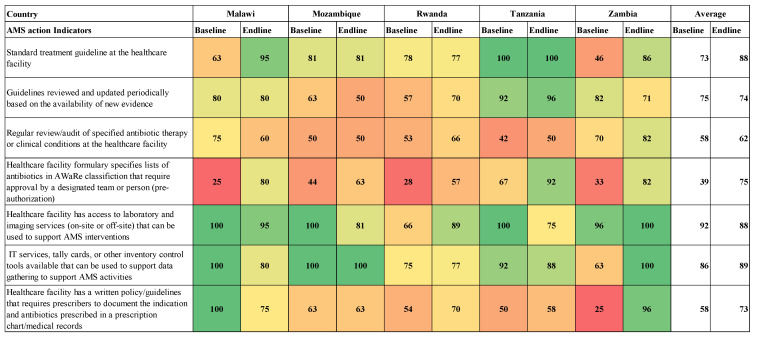
AMS actions across surveyed hospitals.

**Figure 5 antibiotics-14-00266-f005:**
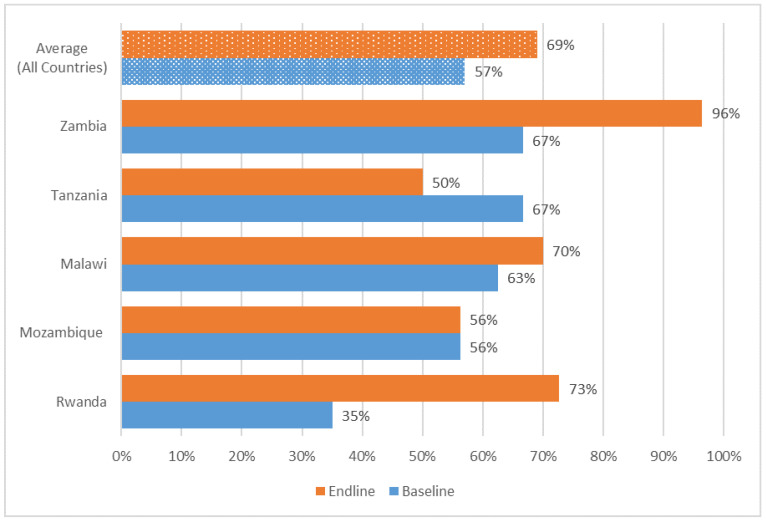
Hospitals offering continuous professional development regarding AMS to staff.

**Figure 6 antibiotics-14-00266-f006:**
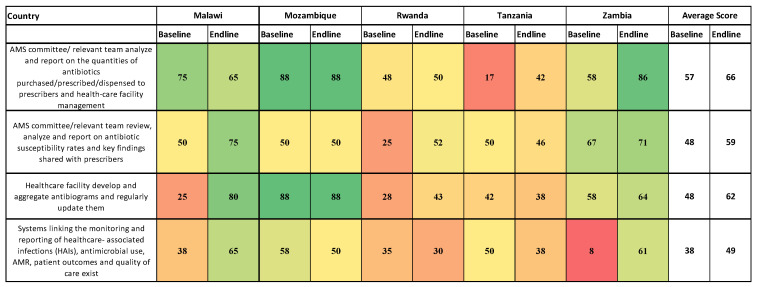
Reporting feedback within healthcare facilities. Green = High Scores Yellow = Moderate Red = Low Score.

**Figure 7 antibiotics-14-00266-f007:**
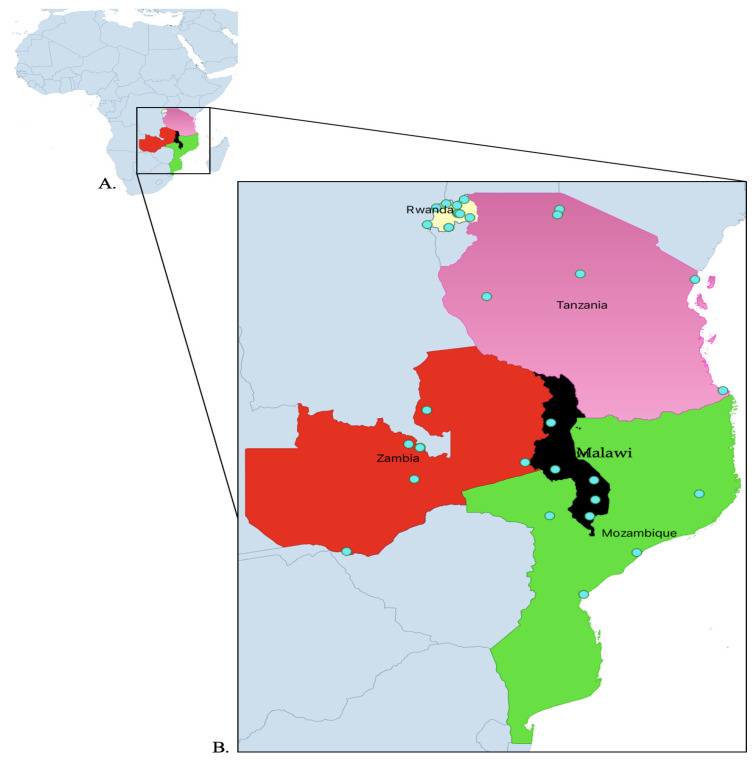
Map indicating surveyed countries (**A**) and distribution of surveyed hospitals (**B**).

**Figure 8 antibiotics-14-00266-f008:**
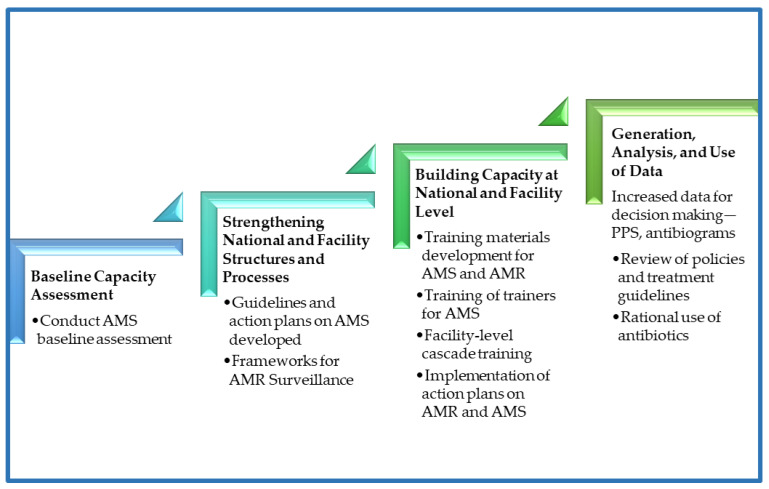
Regional approach to establishment of antimicrobial stewardship programs.

**Figure 9 antibiotics-14-00266-f009:**
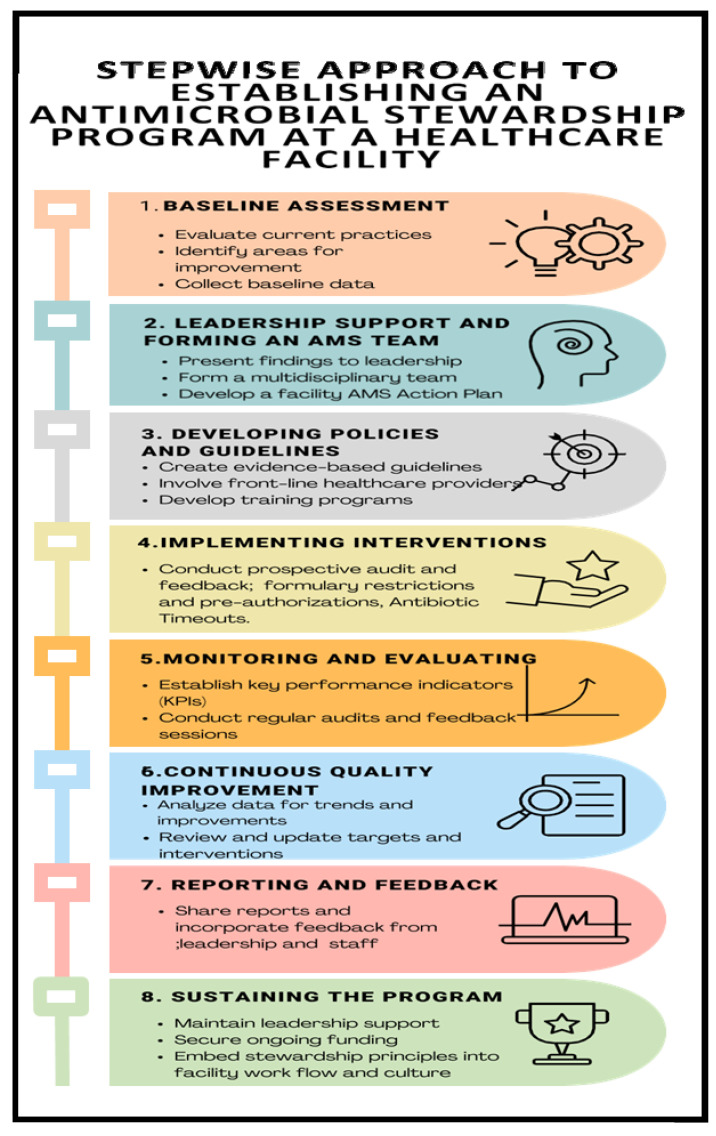
The stepwise approach to establishing an antimicrobial stewardship program at a healthcare facility.

**Table 1 antibiotics-14-00266-t001:** Demographic profiles of assessed hospitals.

Country	Hospital Level	Number of Hospitals
Malawi	Tertiary	2
	Secondary	3
Mozambique	Tertiary	4
Rwanda	Secondary	7
	Tertiary	4
Tanzania	Secondary	6
Zambia	Secondary	2
	Tertiary	4
Total		32

## Data Availability

The original contributions presented in the study are included in the article; further inquiries can be directed to the corresponding author.

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
