# Peer review of "A Regional Approach to Strengthening the Implementation of Sustainable Antimicrobial Stewardship Programs in Five Countries in East, Central, and Southern Africa"

_antibiotics, 2025, doi:10.3390/antibiotics14030266_

Round 1
Reviewer 1 Report
Comments and Suggestions for Authors
Thank you for your submission.
I feel that your paper has demonstrated some significant improvement in antimicrobial stewardship practices in LMIC countries.
There are a few minor items that can be done to improve the manuscript.
- The introduction is a bit lengthy. Please edit or condense the information. More is not always better. For example, the text in line 71 to 104 can be moved elsewhere. If you read the last line from paragraph 2, it transitions nicely to the paragraph starting on line 105.
- Figure 1 column titles need to be re-formatted into something cleaner for the reader.
- Figure 1. Add footers for all acronyms
- Figure 1 and Figure 2. DTC functionally status is listed in both figures. It might be beneficial to change the title on Figure 2 so that the reader gets confused due to reading quickly. Consider retitling figure 2 as "DTC Interventions" or something that better describes the processes.
- Figure 7. I am not sure that we need a map of the hospital locations.
- Discussion. Line 257-318. This section seems longer than it needs to be. Please tighten up the information. I would recommend on making a statement that average scores for each of the assessment showed improvements from baseline. Add a paragraph on areas of greatest improvement so that you don't have to dedicate a paragraph for each assessment. Areas that need further improvement are... Since the study is cross-sectional in design, you can only speculate on specific deficiencies.
- Study limitations: please include common cross-sectional study limitations. With that said, your study is a bit more quasi-experimental in study design. the pre- and post- aspect.
- Table 1. Did you ever look at the data with regard to hospital level, Tertiary vs secondary? This way you can comment on whether hospitals that take care of complex patients is better equipped for AMS.
random edit. Line 298. "enrol" is missing a"L"
Again, great paper. I am excited to see the improvements in AMR. I just think it is longer than it needs to be and there is potential to explore more data comparisons.
Reviewer 2 Report
Comments and Suggestions for Authors
This paper is interesting and well conducted evaluating implementation of Antimicrobial stewardship in Africa using harmonazing regional approach.
I have no particular comments to make on the paper, other than to try to condense the work.In some cases (see table 1 and figure 7), it would be better to keep the table and remove
the figure or the contrary.
Furthermore, I would cut in the Discussion from line 278 to line 287.
They are widely known concepts. I would suggest adding a table in which the acronyms are described (e.g.; DTC =
Drug and Therapeutic Committee, etc.) to make reading an already very long text easier.
Reviewer 3 Report
Comments and Suggestions for Authors
The manuscript presents a well-structured and insightful analysis of AMS programs in five African countries. The study effectively highlights the challenges and successes of implementing these programs using a regional approach, making a valuable contribution to the ongoing efforts to combat antimicrobial resistance in LMICs.
The manuscript study strengths:
- study addresses a critical global health issue, demonstrating how regional coordination can enhance AMS implementation
- study employs a well-defined exploratory cross-sectional design, incorporating WHO guidelines and validated assessment tools
- inclusion of pre- and post-implementation assessments across core AMS elements provides clear evidence of progress
- references are appropriate and include key publications; however, a few recent global (or from other African regions) reports could further strengthen the discussion.
I recommend the publication of manuscript after minor corrections:
- Abstract – Please expand abbreviations upon first use or avoid using them in the abstract (e.g., DTC, WHO, LMICs)
- Line 52 – The phrase “The hope for a renaissance” does not sound scientific; please rephrase it (also, review the entire text to avoid literary or colloquial language)
- Figures 1-5 – Expand all abbreviations in the figure legends. Remember that the figure caption should be placed below the figure
- Table 1 – In the case of Rwanda, “Tertiary” is listed twice. Should these be combined into a single entry?
- Line 443 – Should this be Figure 8 instead of Figure 7?
- Line 455 – Should this be Figure 9 instead of Figures 7 and 8?
- Figure 9 – Improve readability, as it is currently difficult to interpret.
